# The Optimization of Hybrid (Microwave–Conventional) Drying of Sweet Potato Using Response Surface Methodology (RSM)

**DOI:** 10.3390/foods12163003

**Published:** 2023-08-09

**Authors:** Senem Tüfekçi, Sami Gökhan Özkal

**Affiliations:** 1Department of Food Processing, Vocational School of Acıpayam, Pamukkale University, Denizli 20800, Türkiye; stufekci@pau.edu.tr; 2Department of Food Engineering, Faculty of Engineering, Pamukkale University, Denizli 20160, Türkiye

**Keywords:** food drying, hybrid drying, sweet potato, optimization, response surface methodology (RSM), beta-carotene, antioxidants, phenolics

## Abstract

Hybrid microwave–hot air (MW–HA) drying of sweet potatoes was optimized using a face-centered central composite design (FCCCD) with response surface methodology through the desirability function. The independent variables were drying temperature (50–70 °C) and microwave power (0–180 W), while the investigated responses were the drying time (D_t_), the rehydration ratio (RR), the water-holding capacity (WHC), the antioxidant activity change (AA-PC), the total phenolic content change (TPC-PC), and the beta-carotene content change (BC-PC). The main criteria for the optimization of hybrid drying of sweet potatoes was to produce dried potatoes in the shortest drying time with a maximum RR and WHC and with minimum bioactive content (AA, TPC, and BC) loss. The optimum conditions were found to be a drying temperature of 54.36 °C with a microwave power of 101.97 W. At this optimum point, the D_t_, RR, WHC, AA-PC, TPC-PC, and BC-PC were 61.76 min, 3.29, 36.56, 31.03%, −30.50%, and −79.64%, respectively. The results of this study provide new information about the effect of the hybrid drying method (MW–HA) on the rehydration ability and bioactive compounds of sweet potatoes, as well as the optimum values of the process.

## 1. Introduction

Sweet potato is a member of the Convolvulaceae family, which belongs to the genus Ipomoea [1]. Leaves, stems, and roots are the edible parts of sweet potato, and the flesh colors range from white, cream, yellow, and orange to purple [2,3]. Depending on the color of the flesh, sweet potatoes have high concentrations of various bioactive substances such as anthocyanins, beta-carotene, dietary fiber, phenolics, vitamins, and minerals [4]. Some of the health benefits of consuming sweet potatoes are their cardio-protective, immunomodulatory, anticancer, antidiabetic, anti-inflammatory, antimicrobial, anti-obesity, antitumor, and antiulcer effects [5,6].

Following maize, wheat, rice, and potato, the sweet potato is the fifth most widely grown agricultural crop due to its high yield, high efficiency, and resilience in the face of drought [7,8]. With the growing food shortage, suppliers and researchers have begun to focus on vegetables such as sweet potatoes, which are extensively produced yet underutilized in the food industry [5]. Better utilization and commercialization of sweet potatoes can be built on a deeper understanding of how their quality characteristics change in different postharvest processing techniques [9]. One of these techniques is drying, and the dried sweet potatoes can be consumed as snack chips [10] and also ground into flour to be used as a baking ingredient in the production of bread [11], biscuits [12], cakes [13], pasta [14], and noodles [15].

Agricultural crops are dried using the traditional hot air drying method all over the world [16]. This process requires the crop to be exposed to air at high temperatures for an extended period of time, which decreases the overall quality of the end product by damaging the sensory characteristics and nutritional properties of agricultural crops [17] Additionally, case hardening is caused by the movement of soluble materials from the interior of the tissue to its surface, and extreme shrinkage limits the dehydrated products’ rehydration capacity [18,19]. Due to these disadvantages, researchers are working on alternative drying methods such as microwave drying [20]. Microwaves (MW) have the capacity to penetrate dielectric materials and generate heat energy [21]. The generated internal heat creates a vapor pressure in the product, and this moisture is driven to the surface of the material which accelerates the drying process and avoids case hardening [22]. The method of microwave drying has the benefits of high drying rates, improved product quality, elevated energy effectiveness, and space-saving efficiency [23].

Nowadays, in the food drying field, hybrid drying techniques have grown into an important area of research [24]. The strengths and weaknesses of various drying processes have been optimized using hybrid drying techniques [25]. In the process of hybrid microwave–hot air (MW–HA) drying, moisture from the interior of the product is forced to move towards the surface by microwave power, where it is removed by the circulating hot air in the hybrid dryer [26]. High-quality dried products can be manufactured with the lowest energy consumption in the shortest time with MW–HA, depending on the mechanism; this combined method enables heating the interior as well as exterior of the material, which enhances the heat and mass transfer coefficients [27]. Combinational usage of microwave and hot air drying have been utilized for drying different agricultural crops such as apple [28], apricot [29], carrot [30], mushroom [31], nectarine [32], onion [33], pepino [34], persimmon [35], potato [36], sour cherry [37], spinach [38], and tomato [39].

The goal of the optimization of the food drying process is to obtain the best conditions for the system efficiency and dried product quality without additional cost and time [40,41]. For the development, improvement, and optimization of food processes, response surface methodology (RSM) is a practical tool that represents a series of mathematical and statistical approaches [42,43]. RSM analyzes the interactions between the independent variables (input) and dependent variables (response) by creating regression equations [44]. RSM has been widely used in the optimization of drying foods with different methods including the convective air drying of pumpkin seeds [45], the convective–infrared drying of turnip slices [46], the combined microwave–hot air drying of purple cabbage [47], the spray drying of pink guava powder [48], and the vacuum drying of red currants [49].

A limited study has explored the optimization of the microwave–hybrid drying of sweet potatoes, and it focused on drying characteristics such as the effective moisture diffusivity, the activation energy, and drying efficiency of sweet potatoes [50]. This study investigates the effects of the hybrid system on the rehydration characteristics and bioactive compounds; it also performs process optimization along this axis, as an innovation and contribution to the literature.

The influence of drying temperature and microwave energy on the drying time, rehydration ratio, water-holding capacity, antioxidant activity, total phenolic content, and beta-carotene content of sweet potatoes was investigated. This research aims to offer a base of knowledge for enhanced sweet potato drying processing and equipment development by using RSM to optimize the process parameters of the hybrid drying (MW–HA) of sweet potatoes.

## 2. Materials and Methods

### 2.1. Materials and Sample Preparation

Fresh orange-fleshed sweet potatoes (*Ipomoea batatas* [L.] Lam) were purchased from İdeal Tarım Ürünleri Ticareti A.Ş., a fresh vegetable supplier and distributor based in İstanbul, Turkey.

Sweet potato samples were peeled and sliced to 0.4 cm thickness using an electronic vegetable chopper (Moulinex Fresh Express, Moulinex, Ecully, France). In order to maintain a constant shape, a square shape was used to cut the slices into a 2 × 2 cm square. The initial moisture content of sweet potatoes was determined by using a gravimetric method in an oven at 105 °C until a constant weight was reached [51].

### 2.2. Drying Experiments

The drying process was carried out using a microwave oven with 1 m/s air velocity (Siemens HN678G4S1 Built-in Oven, Munich, Germany). This oven permits the simultaneous use of microwave and hot air energy. The oven circulates heated air from a fan and emits microwaves from its top section. The oven combines heated air with microwave power levels of 90, 180, and 360 W. In preliminary studies, 90 and 180 W microwave powers were selected for sweet potato hybrid drying due to the fact that combustion occurred when hot air was coupled with 360 W microwave power.

The drying runs were performed in two parallel runs with three replications each. For each parallel, 16 sweet potato slices were placed on the trays. Total drying time was determined as the duration until the moisture content of sweet potatoes reached 10 ± 0.5%.

### 2.3. Rehydration Characteristics

The experiments for rehydration characteristics (rehydration ratio and water-holding capacity) were replicated three times; two measurements were conducted for each replication, and average values were reported.

#### 2.3.1. Rehydration Ratio

The term “rehydration ratio” refers to the proportion of the mass of the dried sample to the mass of the same sample after it has been rehydrated [52]. The rehydration trials of dried sweet potato samples were conducted at a constant water temperature of 60 °C and an immersion time of one hour. Dried sweet potato samples were placed in distilled water at a sample-to-water ratio of 1:100 (*w*/*w*). Prior to mass measurement, samples were drained for 30 s via a screen to remove the surface water and Equation (1) was used to calculate the rehydration ratio (RR) [53]:(1)$$RR=m_{rt}/m_{r0}$$
where *m_rt_* and *m_r_*_0_ represents the mass of the rehydrated sample and dry sample, respectively.

#### 2.3.2. Water-Holding Capacity

Water-holding capacity (WHC) is the quantity of water held by a known weight of dry materials under given parameters of temperature, soaking time, centrifugation time, and speed [54].

In order to determine the WHC values, rehydrated samples were placed in tubes with a mesh in the center and then centrifuged at 4000 rpm for 10 min at 5 °C. The calculation of WHC values is conducted based on Equation (2) [55]:(2)WHC=[(Mr  Xr – Ms)/(Mr  Xr)]×100
where *Mr* is the mass of the rehydrated sample, *Xr* is the moisture content on a wet basis, and *Ms* is the mass of the removed water from the rehydrated sample after centrifugation.

### 2.4. Bioactive Compounds

The analysis of all bioactive compounds was performed in two parallel runs with three replications.

For the determination of antioxidant activity and total phenolic content of fresh and dried sweet potato slices, an extraction procedure was exactly carried out as described by Özgören et al. [56].

#### 2.4.1. Antioxidant Activity (AA)

AA was measured by using the 2.2-diphenyl-1-picrylhydrazyl (DPPH) method [57,58]. The reaction between 150 μL of extract solution and 2850 μL of DPPH solution was carried out for 1 h in darkness. Then, the absorbance was read at 515 nm wavelength. The antioxidant activity results were obtained as mmol Trolox equivalent (TE)/g dry matter and converted to a percentage change (AA-PC) for use as a response in optimization (Equation (3)):(3)PC %=FvIv−1×100
where *PC* (%) is the percentage change, *Fv* is the final value, and *Iv* is the initial value.

#### 2.4.2. Total Phenolic Content (TPC)

Slight modifications were made to the Folin–Ciocalteu method to determine the TPC of fresh and dried sweet potatoes [59]. A 300 mL volume of extract solution was mixed with 1500 mL of 1 N Folin–Ciocalteu (1:10 *v*/*v*) reagent and 1200 mL of 7.5% (*w*/*v*) Na_2_CO_3_ solution. Then, the mixture was stirred with a vortex, and it was kept at room temperature in darkness for two hours. The absorbance of the incubated mixture was measured at 760 nm. The results were obtained in mg of gallic acid equivalent (GAE)/100 g dry matter, and then the percent change (TPC-PC) was determined in order to be used as a response in the optimization process (Equation (3)).

#### 2.4.3. Beta-Carotene Content

Beta-carotene content was determined by modifying Demiray et al.’s [60] methodology using a HPLC instrument (Shimadzu LC-20AD, Shimadzu Corporation, Kyoto, Japan).

For the extraction of beta-carotene, fresh and dried sweet potato samples weighing 0.5 g were homogenized for one minute in a polypropylene centrifuge tube using a 20 mL ethanol–hexane solution (4:3 *v*/*v*) that contained 1% butylated hydroxytoluene (*w*/*v*). The homogenized samples were subjected to centrifugation for a duration of 15 min at 4 °C and a speed of 9000 rpm. Then, the supernatants were transferred to amber-colored bottles. Prior to HPLC injection, the supernatants were filtered using 0.45 μm membrane filters.

The HPLC system was previously described in [61]. The mobile phase flow rate was set at 0.25 mL/min, and the proportions of acetonitrile, methanol, dichloromethane, and hexane in the mobile phase were in a volumetric ratio of 40:20:20:20. The duration of the assay, the injection volume, and the wavelength of detection were 20 min, 20 µL, and 445 nm, respectively. The beta-carotene content of fresh and dried sweet potato samples was calculated from peak areas by utilizing a standard curve of beta-carotene (*R*^2^ > 0.99).

The percent change in beta-carotene (BC-PC) was also estimated on a dry basis for the optimization calculations (Equation (3)).

### 2.5. Experimental Design, Optimization, and Statistical Analysis

Response surface methodology (RSM) was used to optimize the hybrid drying (MW–HA) of sweet potatoes. Drying temperature and microwave power were selected as independent variables. Coded and uncoded (actual) levels of independent variables are listed in Table 1. Drying time, rehydration ratio, water-holding capacity, total color change, antioxidant activity, total phenolic content, and beta-carotene content were chosen as responses.

The experiments were planned using a face-centered central composite design (FCCCD) in randomized order (Table 2). The experimental design consisted of 13 experimental runs with five replicates at the center point.

The CCD is an effective design that works successfully for sequential experimentation and provides adequate information to test the lack of fit without requiring a large number of design points. A quadratic surface can be fitted using CDD, which is frequently effective for process optimization [62]. In the present study, a CCD was employed to determine the optimum conditions for hybrid drying of sweet potatoes.

In RSM, the model equation is the complete second-order equation, and the experimental data obtained from the FCCCD experiments can be represented in the form of the following equation (Equation (4)) [63,64]:(4)$$Y=b_{o}+\sum_{i=1}^{n}b_{i}x_{i}+\sum_{i=1}^{n}b_{ii}x_{i}^{2}+\sum_{i=1}^{n-1}\sum_{j=i+1}^{n}b_{ij}x_{i}x_{j}+\varepsilon$$
where *Y* is the dependent variable (desired value of response), *b_o_* is the constant, and *b_i_*, *b_ii_*, and *b_ij_* are linear coefficients, quadratic coefficient, and interaction effect coefficients, respectively. The terms *x_i_* and *x_ij_* are the coded levels of the independent variables; *n* is the number of independent variables; and *ε* is the random error term.

The optimization and statistical studies were performed utilizing Design Expert Software, version 12.0 (Stat-Ease Inc., Minneapolis, MN, USA). The statistical analysis was conducted using significance levels of *p* = 0.01 and *p* = 0.05.

In the approach of numerical optimization, the desirability function was used to determine the optimum levels of independent variables. The desirability function is a composite function that describes how well-fitting the responses are at a given level of independent variables and is calculated by Equation (5) [65]:(5)D=d1×d2×d3………×dn 1n
where *D* is the overall desirability, *d* is the individual desirability, and *n* is the number of responses. The software of the program searches for variable values that can produce the optimum value of the desirability function, which ranges from 0 to 1.

The significance of the model was tested by performing an ANOVA test by estimating the F-ratio, which is the ratio between the regression mean square and the error mean square. *R*^2^, *Adjusted R*^2^, and *Predicted R*^2^ are the coefficients of determination that also describe how well the model performed (Equations (6)–(8)) [66]:(6)$$R^{2}=1-\frac{{SS}_{residual}}{{SS}_{total}}$$
(7)$$Adjusted\;R^{2}=1-\frac{{SS}_{residual}/{DF}_{residual}}{{SS}_{total}/{DF}_{total}}$$
(8)$$Predicted\;R^{2}=1-\frac{{\sum_{1}^{n}{(y}_{i}-{\bar{y}}_{i\;})}^{2}}{{SS}_{total}}$$
where *SS_residual_* is the sum of squares of the differences between the predicted and actual values, *SS_total_* is the sum of squares of the differences between the predicted and average of actual values, *y_i_* is the *i*th value of the variable to be predicted, and ${\bar{y}}_{i\;}$ is the predicted value corresponding to *y_i_*.

## 3. Results and Discussion

Findings from different repetitions of drying runs are displayed in Table 2. The analysis of variance (ANOVA) was used to examine the significance level of possible regression models and independent variables on each response (Table 3, Table 4, Table 5 and Table 6). To visualize the single and combining effects of independent variables on all responses, three-dimensional response surface contour plots (3D-RSCP) were generated for every response (Figure 1, Figure 2, Figure 3 and Figure 4). The equations to be used in calculating the estimated values of the responses were created in terms of coded factors (Equations (9)–(14)).

### 3.1. Drying Time (D_t_)

It was observed that the D_t_ shortened with rising drying temperatures and microwave power (Figure 1), and the shortening caused by the microwave power in the D_t_ decreased with rising air temperatures (Table 2). The savings in drying time (%) were computed by dividing the drying time reduction in hybrid drying relative to the total drying time of convection drying [67]. Compared to drying experiments conducted with 50, 60, and 70 °C hot air alone, drying conditions where hot air was combined with 90 and 180 W microwave power reduced total drying time by 59.89% and 71.19%, 51.80% and 65.57%, and 44.00% and 63.00%, respectively. It is also reported in other studies that the combination of microwave power with hot air can significantly shorten the process time for lemon slices [68], onion slices [33], sour cherries [39], and sweet potatoes [50].

The quadratic model was found to be the best suitable model, with a significant F value and an insignificant lack of fit (*p* < 0.01). The *Predicted R*^2^ (0.9557) is in agreement with the *Adjusted R*^2^ (0.9890), with a difference of less than 0.2. As indicated in Table 3, drying temperature (A), microwave power (B), and drying temperature–microwave power interaction (AB) are significant model terms.

The drying time regression equation relating coded levels of process parameters was found as (Equation (9)):(9)$$\frac{1}{D_{t}}=0.0169+0.0026A+0.0078B+0.0008AB-0.0007A^{2}-0.0007B^{2}$$

### 3.2. Rehydration Ratio (RR) and Water-Holding Capacity (WHC)

In view of the importance of consumer acceptance, it is highly suggested that dried products have adequate rehydration power [49]. The rehydration ratio of dried sweet potatoes increased with increasing microwave power for hybrid drying at 50 °C. However, it was observed that the rehydration ratio of sweet potatoes dried at 60 and 70 °C were lower (Table 2) (Figure 2a). Similar behavior occurred in the rehydration of microwave–convective dried pomegranate arils at 70 °C, and this situation was explained as a result of increased drying temperatures causing significant tissue collapse and shrinkage, which in turn led to a reduction in the rehydration capacity [69]. In addition, the rehydration rates of hybrid dried (MW–HA) purple cabbage first increased and then decreased with rising drying temperature and microwave power [47]. The increase in the rehydration rate of the samples dried at 50 °C + 90 W and 50 °C + 180 W combinations is thought to be due to the significant decline in the drying time, similar to the hybrid (MW–HA) dried kiwifruit samples [70]. It was also observed that the water-holding capacity of the samples dried under the conditions of 180 W microwave power and hot air was combined lower than the other samples dried at the same temperature (Table 2). This is a result of the increased microwave power destroying the cell wall and other supporting structures [71].

The quadratic model was found to be the most suitable model for both rehydration ratio and water-holding capacity. While drying temperature (A), drying temperature–microwave power interaction (AB), and quadratic effect of microwave power (B^2^) are significant terms for rehydration ratio, microwave power (B) and drying temperature–microwave power interaction (AB) were significant for water-holding capacity (Table 4). Considering the drying temperature–microwave power interaction (AB) was a significant term for both of these responses (RR and WHC), it can be concluded that the hybrid drying system is effective on the rehydration properties of sweet potatoes.

The regression equations for rehydration ratio (Equation (10)) and water-holding capacity (Equation (11)) in terms of coded values are given below:(10)$$RR=3.23-0.0825\;A+0.0069\;B-0.1108\;AB+0.0170\;A^{2}-0.0953\;B^{2}$$
(11)$$WHC=37.99+1.59\;A-3.90\;B-2.83\;AB-0.5783\;A^{2}-2.54\;B^{2}$$

### 3.3. Antioxidant Activity (AA-PC) and Total Phenolic Content Changes (TPC-PC)

At the end of the drying process, the antioxidant activity of all samples increased. The antioxidant capacity increases with increasing microwave power (Figure 3a). The highest values in the increase of antioxidant activity were realized in conditions where hot air was combined with 180 W microwave power (Table 2). This pattern of behavior could be attributable to the fact that the drying process was carried out at relatively low temperatures, which requires longer drying times and may have contributed to the reduction in antioxidant capability [72]. The products of the Maillard reaction have also been linked to an increase in antioxidant capability upon drying [73].

As given in Table 2, the total phenolic content loss decreased with increasing microwave power and drying temperature (Figure 3b). The loss of total phenolic content in conventional drying was greater than the hybrid drying as a result of extended exposure to oxygen and thermal degradation of the phenolics [37]. On the other hand, as in this study, rapid heating in hybrid drying can deactivate oxidative enzymes and improve the retention of phenolic components [74].

Drying temperature (A), microwave power (B), drying temperature–microwave power interaction (AB), quadratic effect of drying temperature (A^2^), and quadratic effect of microwave power (B^2^) were significant model terms for antioxidant activity change in the fitted quadratic model. When the ANOVA analysis is examined, it can be seen that hybrid drying has a significant effect on the alteration in the antioxidant activity of sweet potatoes (Table 5).

The quadratic model was the best-fitted model with experimental data of total phenolic content change (%) and microwave power (B), drying temperature–microwave power interaction (AB), quadratic effect of drying temperature (A^2^), and quadratic effect of microwave power (B^2^) were significant model terms (Table 5)

The regression equations for antioxidant activity change (%) (Equation (12)) and total phenolic content change (%) (Equation (13)) in terms of coded levels of process factors are given below:(12)$$AA-PC=30.01-2.76\;A+7.37\;B-15.49\;AB-8.69\;A^{2}+4.83\;B^{2}$$
(13)$$TP-PC=-32.99+1.18\;A+11.56\;B-5.30\;AB+4.17\;A^{2}-6.26\;B^{2}$$

### 3.4. Beta-Carotene Content Change (BC-PC)

Beta-carotene content decreased with the drying process, and the highest losses in beta-carotene content occurred in conditions where 50 and 70 °C hot air were combined with 180 W microwave power (Table 2). Similar declines were reported for the drying process of apricot, and this situation was explained by the fact that beta-carotene is a compound that is prone to degradation at high temperatures and that degradation is accelerated by the microwave effect [75]. Contrary to this situation, the loss of beta-carotene did not change significantly in the samples dried only with hot air. The highest loss occurred in the sample dried at 50 °C in comparison to 60 and 70 °C due to the long drying time (Figure 4) (Table 3). For African eggplant fruit, the decrease in beta-carotene retention with a longer drying time is also mentioned [76].

For the beta-carotene content change values of sweet potatoes, the most suitable model was found as quadratic model with significant a F value and an insignificant lack of fit (*p* < 0.05). For this response, the quadratic effect of drying temperature (A^2^), microwave power (B), and quadratic effect of microwave power (B^2^) were significant model terms (Table 6). As can be observed in Figure 4, microwave power is the dominant process parameter affecting beta-carotene content change in hybrid drying.

The beta-carotene change regression equation relating coded levels of process parameters was found as (Equation (14)).
(14)$$BC-PC=-76.80+0.3633\;A-1.16\;B-0.4150\;AB-2.01\;A^{2}+3.95\;B^{2}$$

### 3.5. Numerical Optimization of the Hybrid Drying Process

The expected objectives for each independent variable and response are detailed in Table 7. All the process parameters and the responses were assigned an equal importance weight of three. While the major criterion was to produce dried sweet potatoes with the maximum bioactive component preservation and rehydration qualities using hybrid drying (MW–HA), the targets of the responses were defined within the range of the lower and upper limits.

For the optimization of the hybrid drying process of sweet potatoes, the numerical optimization method was used, and the objective was to find a point that provided the greatest possible increase in the desirability function. Figure 5 shows the desirability values of independent and dependent variables, as well as the combined desirability of the optimization process. Since they are set to be in-range in the optimization, the desirability function of drying temperature and microwave power is equal to one. The desirability function value of combined optimization was 0.645.

The optimum values of drying process variables were calculated as a drying temperature of 54.36 °C and microwave power of 101.97 W. The values of drying time, rehydration ratio, water-holding capacity, antioxidant activity change, total phenolic activity change, and beta-carotene content change responses were predicted as 61.76 min, 3.29, 36.56, 31.03%, −30.50%, and −79.64%, respectively (Table 6).

## 4. Conclusions

Hybrid drying (MW–HA) of sweet potatoes was investigated, and to optimize the process parameters of hybrid drying of sweet potatoes, response surface methodology was used. Drying air temperature and microwave power were independent factors, and drying time (D_t_), rehydration ratio (RR), water-holding capacity (WHC), antioxidant activity change (AA-PC), total phenolic content change (TPC-PC), and beta-carotene content change (BC-PC) were selected as responses. The drying time is shortened with hybrid drying, and the total phenolic and beta-carotene content of sweet potatoes was reduced. On the other hand, antioxidant activity increased with hybrid drying, according to the formation of Millard reaction. All responses were fitted to a quadratic model, and the AB factor, which represented the simultaneous effect of hot air and microwave power, is significant for D_t_, RR, WHC, AA-PC, and TPC-PC responses (*p* < 0.05). Drying temperature of 54.36 °C and microwave power of 101.97 W were optimized conditions for sweet potato hybrid drying, and the predicted Dt, RR, WHC, AA-PC, TPC-PC, and BC-PC were 61.76 min, 3.29, 36.56, 31.03%, −30.50%, and −79.64%, respectively. The desirability function for the combination was 0.645.

According to the study findings, hybrid drying (MW–HA) has the potential to be a practical and profitable approach for drying food. Microwave–hot air hybrid drying technologies have been extensively researched through experimental, theoretical, and numerical studies. However, there is almost no information about large-capacity industrial applications. Future studies can be focused in two directions: the energy efficiency of hybrid drying systems and the design of medium- and large-capacity microwave hot air hybrid dryers.

## Figures and Tables

**Figure 1 foods-12-03003-f001:**
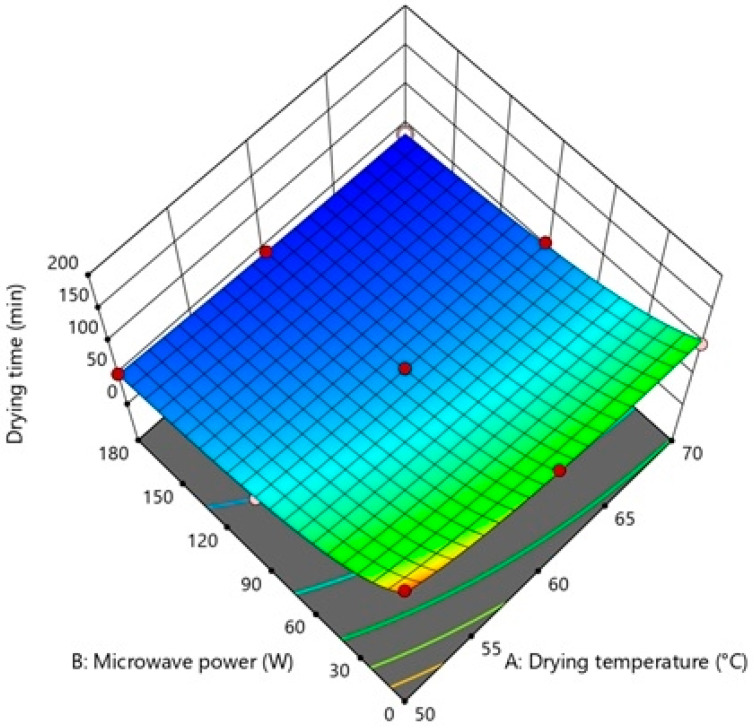
3D-RSCP showing the effects of drying temperature, microwave power, and their mutual interactions on drying time.

**Figure 2 foods-12-03003-f002:**
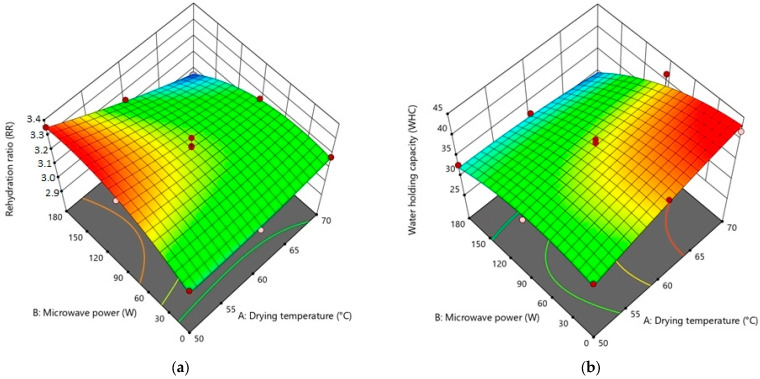
3D-RSCP showing the effects of drying temperature, microwave power, and their mutual interactions on (**a**) rehydration ratio (**b**) water-holding capacity.

**Figure 3 foods-12-03003-f003:**
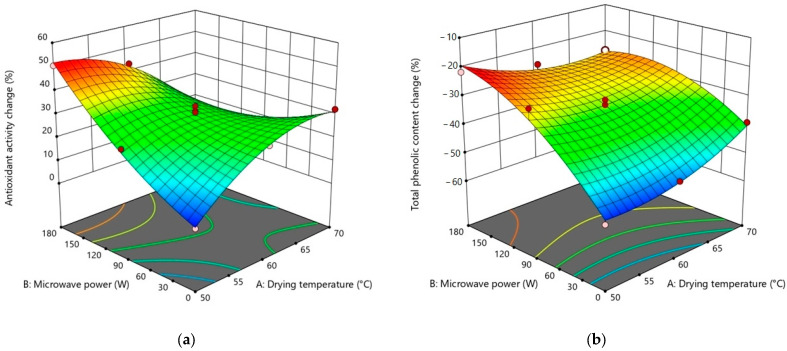
3D-RSCP showing the effects of drying temperature, microwave power, and their mutual interactions on (**a**) antioxidant activity (**b**) total phenolic content.

**Figure 4 foods-12-03003-f004:**
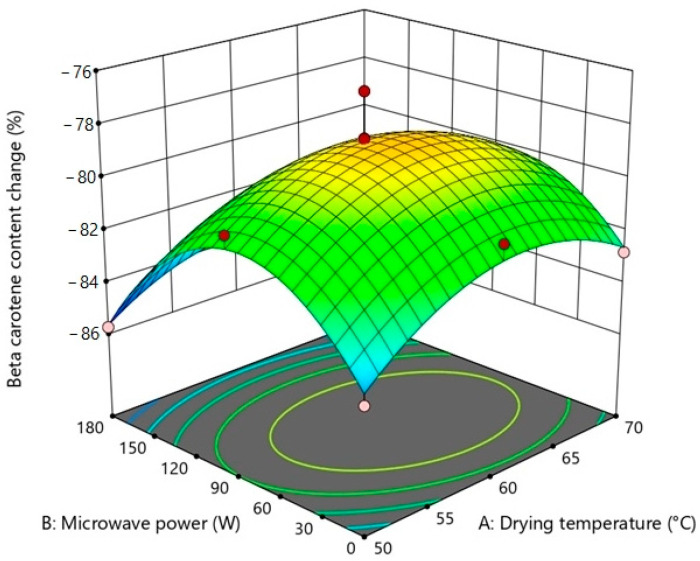
3D-RSCP showing the effects of drying temperature, microwave power, and their mutual interactions on beta-carotene content.

**Figure 5 foods-12-03003-f005:**
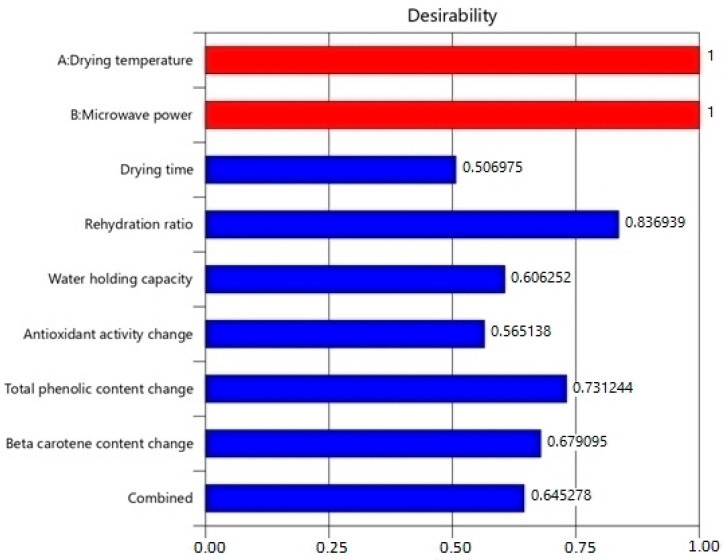
Desirability values of independent, dependent variables, and combined optimization.

**Table 1 foods-12-03003-t001:** Independent variables.

Factor Code	Name	Unit	Coded/Actual Levels		
			−1	0	1
A	Drying temperature	°C	50	60	70
B	Microwave power	W	0	90	180

**Table 2 foods-12-03003-t002:** FCCCD experimental design and observed values for hybrid drying of sweet potatoes independent variables.

Run	Independent Variables		Responses					
	DT (°C)	MW (W)	D_t_ (min)	RR	WHC	AA-PC (%)	TPC-PC (%)	BC-PC (%)
4	50	0	177	3.14	35.01	5.58	−54.61	−84.61
13	50	90	71	3.30	34.50	25.74	−26.27	−80.47
7	50	180	51	3.35	32.90	50.59	−21.65	−85.74
9	60	0	122	3.09	40.06	27.15	−50.45	−80.78
1	60	90	58	3.24	38.74	26.48	−34.47	−79.89
2	60	90	61	3.30	35.52	26.69	−32.86	−79.84
3	60	90	57	3.19	39.58	31.49	−33.81	−78.49
5	60	90	60	3.24	36.91	33.67	−34.91	−76.75
6	60	90	58	3.24	37.90	30.92	−31.11	−78.51
12	60	180	42	3.15	32.18	43.33	−25.82	−83.83
8	70	0	100	3.18	41.84	17.72	−39.03	−82.80
11	70	90	56	3.17	41.65	15.35	−29.14	−80.25
10	70	180	37	2.95	28.44	32.31	−27.28	−85.59

**Table 3 foods-12-03003-t003:** ANOVA for drying time response.

Source	Sum of Squares (SSS)	Df	Mean Square	F Value	*p* Value	
**Transform: Inverse**						
**Model**	0.0004	5	0.0001	216.89	<0.0001 **	Significant
A-Drying temperature	0.0000	1	0.0000	106.66	<0.0001 **	
B-Microwave power	0.0004	1	0.0004	958.81	<0.0001 **	
AB	2.355 × 10^−6^	1	2.355 × 10^−6^	6.24	0.0411 *	
A^2^	1.553 × 10^−6^	1	1.553 × 10^−6^	4.11	0.0821	
B^2^	1.422 × 10^−6^	1	1.422 × 10^−6^	3.77	0.0934	
**Residual**	2.642 × 10^−6^	7	3.774 × 10^−7^			
Lack of Fit	1.752 × 10^−6^	3	5.841 × 10^−7^	2.63	0.1870	Not significant
Pure Error	8.898 × 10^−7^	4	2.225 × 10^−7^			
**Cor. Total**	0.0004	12				

* *p* < 0.05, ** *p* < 0.01. *R*^2^ = 0.9936, Adjusted *R*^2^ = 0.9890, Predicted *R*^2^ = 0.9957.

**Table 4 foods-12-03003-t004:** ANOVA for rehydration ratio and water-holding capacity responses.

	Response: Rehydration Ratio					
Source	SSS	Df	Mean Square	F Value	*p* Value	
**Transform: None**						
**Model**	0.1166	5	0.0233	16.50	0.0009 **	Significant
A-Drying temperature	0.0408	1	0.0408	28.90	0.0010 **	
B-Microwave power	0.0003	1	0.0003	0.2002	0.6681	
AB	0.0492	1	0.0492	3.79	0.0006 **	
A^2^	0.0008	1	0.0008	0.5643	0.4770	
B^2^	0.0251	1	0.0251	17.76	0.0040 **	
**Residual**	0.0099	7	0.0014			
Lack of Fit	0.0023	3	0.0008	0.4152	0.7519	Not significant
Pure Error	0.0075	4	0.0019			
**Cor. Total**	0.1265	12				
	**Response: Water-Holding Capacity**					
**Source**	**SSS**	**Df**	**Mean Square**	**F value**	***p* value**	
**Transform: None**						
**Model**	163.74	5	32.75	9.38	0.0052 **	Significant
A-Drying temperature	15.12	1	15.12	4.33	0.0759	
B-Microwave power	91.17	1	91.17	26.13	0.0014 **	
AB	31.94	1	31.94	9.15	0.0192 *	
A^2^	0.9237	1	0.9237	0.2647	0.6228	
B^2^	17.79	1	17.79	5.10	0.0585	
**Residual**	24.43	7	3.49			
Lack of Fit	14.41	3	4.80	1.92	0.2684	Not significant
Pure Error	10.02	4	2.51			
**Cor. Total**	188.16	12				

* *p* < 0.05, ** *p* < 0.01. *R*^2^ = 0.9218, Adjusted *R*^2^ = 0.8659, Predicted *R*^2^ = 0.7402. *R*^2^ = 0.8702, Adjusted *R*^2^ = 0.7775, Predicted *R*^2^ = 0.1880.

**Table 5 foods-12-03003-t005:** ANOVA for antioxidant activity change and total phenolic content change responses.

	Response: Antioxidant Activity Change					
Source	SSS	Df	Mean Square	F Value	*p* Value	
**Transform: None**						
**Model**	1547.29	5	309.46	46.12	<0.0001 **	Significant
A-Drying temperature	45.54	1	45.54	6.79	0.0352 *	
B-Microwave power	326.05	1	326.05	48.59	0.0002 **	
AB	960.07	1	960.07	143.08	<0.0001 **	
A^2^	208.15	1	208.15	31.02	0.0008 **	
B^2^	64.40	1	64.40	9.60	0.0174 *	
**Residual**	46.97	7	6.71			
Lack of Fit	7.21	3	2.40	0.2416	0.8636	Not significant
Pure Error	39.77	4	9.94			
**Cor. Total**	1594.26	12				
	**Response: Total Phenolic Content Change**					
**Source**	**SSS**	**df**	**Mean Square**	**F value**	***p* value**	
**Transform: None**						
**Model**	1040.77	5	208.15	37.77	<0.0001 **	Significant
A-Drying temperature	8.35	1	8.35	1.52	0.2580	
B-Microwave power	801.34	1	801.34	145.40	<0.0001 **	
AB	112.47	1	112.47	20.41	0.0027 **	
A^2^	47.92	1	47.92	8.70	0.0214 *	
B^2^	108.39	1	108.39	19.67	0.0030 **	
**Residual**	38.58	7	5.51			
Lack of Fit	29.46	3	9.82	4.30	0.0962	Not significant
Pure Error	9.12	4	2.28			
**Cor. Total**	1079.35	12				

* *p* < 0.05, ** *p* < 0.01. *R*^2^ = 0.9705, Adjusted *R*^2^ = 0.9495, Predicted *R*^2^ = 0.9195. *R*^2^ = 0.9643, Adjusted *R*^2^ = 0.9387, Predicted *R*^2^ = 0.7323.

**Table 6 foods-12-03003-t006:** ANOVA for beta-carotene change response.

Source	SSS	Df	Mean Square	F Value	*p* Value	
**Transform: Inverse**						
**Model**	92.72	5	18.54	17.21	0.0008 **	Significant
A-Drying temperature	0.7921	1	0.7921	0.7353	0.4196	
B-Microwave power	8.10	1	8.10	7.52	0.0289 *	
AB	0.6889	1	0.6889	0.6395	0.4502	
A^2^	11.15	1	11.15	10.36	0.0147 *	
B^2^	43.19	1	43.19	40.10	0.0004 **	
**Residual**	7.54	7	1.08			
Lack of Fit	0.9421	3	0.3140	0.1904	0.8979	Not significant
Pure Error	6.60	4	1.65			
**Cor. Total**	100.26	12				

* *p* < 0.05, ** *p* < 0.01. *R*^2^ = 0.9248, Adjusted *R*^2^ = 0.8711, Predicted *R*^2^ = 0.8201.

**Table 7 foods-12-03003-t007:** The constraints and results of optimizing the hybrid drying process of sweet potatoes.

Name	Goal	Lower Limit	Upper Limit	Importance Level	Predicted
**A: DT**	In range	50	70	3	54.36 °C
**B: MW**	In range	0	180	3	101.97 W
**D_t_**	Minimize	177	37	3	61.76 min
**RR**	Maximize	2.95	3.35	3	3.29
**WHC**	Maximize	28.44	41.84	3	36.56
**AA-PC**	Maximize	5.58	50.59	3	31.03%
**TPC-PC**	Maximize	−54.61	−21.65	3	−30.50%
**BC-PC**	Maximize	−85.74	−76.75	3	−79.64%

## Data Availability

All relevant data of this study are presented in the paper. Contact the corresponding author with any further questions.

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
