# Peer review of "The Optimization of Hybrid (Microwave–Conventional) Drying of Sweet Potato Using Response Surface Methodology (RSM)"

_foods, 2023, doi:10.3390/foods12163003_

Round 1

Reviewer 1 Report

The authors investigated the influence of hybrid (microwave-conventional) drying on drying time, rehydration ratio, water holding capacity, antioxidant activity, total phenolic content, beta-carotene content of sweet potatoes.

The results are interesting and important. However, some issues should be clarified and corrected.

The objective and novelty should be indicated in the abstract more clearly.

The novelty of the present study against the background of available literature should be indicated in detail in the Introduction.

Materials should be described in more detail. What was the sample size? How many fresh sweet potatoes were obtained? How many slices?

The number of repetitions should be given for each type of measurement.

A separate Statistical analysis subsection should be added.

Future research should be indicated in detail.

Reviewer 2 Report

The paper is clearly written, nicely organized.

Comment for revision:

- the plots in Figure 1 and Figure 2 should be rotated so the points and the surface can be clearly seen.

- in the subsection 2.5. Experimental design, more details about the RSM should be provided.

 Minor editing of English language required

Reviewer 3 Report

This manuscript has been written well, but there is some corrections need to be done:

1) Line 110 - Why solid/liquid ratio of dried apple is required in this study?

2) Line 196 - 197 - Statement ' Compared to drying experiments conducted with 50, 60 and 70C hot air alone ...'. Did the author done the drying using hot air only? Where is the results that the author compared with the hybrid process?

3) Line 258 - I assume there is a spelling error for the term 'loos', or loss'?

4) Line 319 - 323- Optimization values have been predicted, but there is no validation has been conducted? Why the authors did not conducting the validation? Percentage of error can be obtained between predicted and validated values. 

Reviewer 4 Report

Optimization of Hybrid (Microwave-Conventional) Drying of Sweet Potato Using Response Surface Methodology (RSM) with the manuscript no. foods-2505093 has been reviewed. Following listed serious issues are recommended first to considered by the authors.

Table 2 (Column 1) – as Run 1 to 6 have the same independent variables (60 oC and 90 W), the theoretical values of dependent variables (RR, WHC, AA-PC, TPC-PC and BC-PC) must be the same; so, can you please explain such differences in results (RR – 3.19 to 3.3; WHC – 35.52 to 39.58; AA-PC – 26.48 to 33.67; TPC-AA – 31.11 to 34.91; BC-PC – 76.75 to 79.89) for the same drying conditions?

Section 3.1 (Line 198): “Compared to drying experiments conducted with 50, 60 196 and 70 °C hot air alone, drying conditions where hot air was combined with 90 and 180 197 W microwave power reduced total drying time as 59.89 % and 71.19 %, 51.64 % and 65.57 %, 44.00 % and 63.00 %, respectively.” For the drying time reduction through the use of MW at 60 oC hot-air temperature, which Dt value from Table 2 was used to calculate the drying time reduction (51.64 % and 65.57 %)?

Table 7. ANOVA for beta carotene change response. – I don’t think that Table 7 is about beta carotene change response?

Table 7: From where these predicted values of dependent and independent variables were calculated / obtained?

Needs extensive revision in English writing; for example, no unit for a value; start a new sentence after a comma instead of a full-stop; "where" or "Where" instead of "where," after an equation; ml instead of mL; non-italic p; -79,64 % instead -79.64% etc. 

Round 2

Reviewer 1 Report

The manuscript has been improved

Reviewer 4 Report

The authors have addressed all the issues raised.

None.